# Adipocyte Specific HO-1 Gene Therapy Is Effective in Antioxidant Treatment of Insulin Resistance and Vascular Function in an Obese Mice Model

**DOI:** 10.3390/antiox9010040

**Published:** 2020-01-01

**Authors:** Shailendra P. Singh, Menachem Greenberg, Yosef Glick, Lars Bellner, Gaia Favero, Rita Rezzani, Luigi Fabrizio Rodella, Kevin Agostinucci, Joseph I. Shapiro, Nader G. Abraham

**Affiliations:** 1Departments of Medicine and Pharmacology, New York Medical College, Valhalla, NY 10595, USA; spbiotech2004@gmail.com (S.P.S.); mgreenbe20@nymc.edu (M.G.); yglick2@nymc.edu (Y.G.); Lars_Bellner@nymc.edu (L.B.); kagostin@student.nymc.edu (K.A.); 2Anatomy and Physiopathology Division, Department of Clinical and Experimental Sciences, University of Brescia, 25123 Brescia, Italy; gaia.favero@med.unibs.it (G.F.); rezzani@med.unibs.it (R.R.); rodella@med.unibs.it (L.F.R.); 3Interdepartmental University Center of Research “Adaption and Regeneration of Tissues and Organs-(ARTO)”, University of Brescia, 25123 Brescia, Italy; 4Departments of Cardiology and Internal Medicine, Joan C. Edwards School of Medicine, Marshall University, Huntington, WV 25701, USA; ShapiroJ@marshall.edu

**Keywords:** antioxidant, gene-HO-1, adiponectin, adipocytes, PGC1-α, hyperglycemia, hypertension

## Abstract

Obesity is a risk factor for vascular dysfunction and insulin resistance. The study aim was to demonstrate that adipocyte-specific HO-1 (heme oxygenase-1) gene therapy is a therapeutic approach for preventing the development of obesity-induced metabolic disease in an obese-mice model. Specific expression of HO-1 in adipose tissue was achieved by using a lentiviral vector expressing HO-1 under the control of the adiponectin vector (Lnv-adipo-HO-1). Mice fed a high-fat diet (HFD) developed adipocyte hypertrophy, fibrosis, decreased mitochondrial respiration, increased levels of inflammatory adipokines, insulin resistance, vascular dysfunction, and impaired heart mitochondrial signaling. These detrimental effects were prevented by the selective expression of HO-1 in adipocytes. Lnv-adipo-HO-1-transfected mice on a HFD display increased cellular respiration, increased oxygen consumption, increased mitochondrial function, and decreased adipocyte size. Moreover, RNA arrays confirmed that targeting adipocytes with HO-1 overrides the genetic susceptibility of adiposopathy and correlated with restoration of the expression of anti-inflammatory, thermogenic, and mitochondrial genes. Our data demonstrate that HO-1 gene therapy improved adipose tissue function and had positive impact on distal organs, suggesting that specific targeting of HO-1 gene therapy is an attractive therapeutic approach for improving insulin sensitivity, metabolic activity, and vascular function in obesity.

## 1. Introduction

Obesity has become highly prevalent in the past decade worldwide, affecting all age groups and populations [1,2,3]. It is estimated that approximately 35% of the general population in high-income countries are overweight, and 30% are obese [2]. Obesity is a risk factor for type 2 diabetes, coronary heart disease, chronic kidney disease, hypertension, and high cholesterol levels [3]. In the United States, approximately 70% of adults and 32% of children are currently either overweight or obese, and the percent of obese patients continues to rise every year [4]. Obesity is associated with inflammation of the heart, often leading to myocardial infarction, reduction in left ventricular function, and reduced ejection fraction [5,6]. White fat, whether epicardial or visceral, is considered a major source of inflammation [7,8,9] and leads to a reduction in mitochondrial thermogenesis and biogenesis signaling [10,11,12].

In addition to increasing ectopic fat around the heart, obesity leads to insulin resistance in humans [13]. Visceral fat and insulin resistance are related to metabolic disease, as a result of a decrease of HO-1 show elevation of reactive oxygen species (ROS) [14,15,16,17,18] and diminished levels of the antioxidant gene heme oxygenase-1 (HO-1). Low levels of visceral adipose tissues HO-1 are correlated with increased hip-to-waist ratio and insulin resistance [19]. Importantly, reduced levels of HO-1 are associated with reduced antioxidative properties and insulin resistance associated with obesity [11,20,21]. In fact, humans and mice lacking HO-1 suffer from severe organ damage due to major increases in ROS [14,15,16,17,18]. ROS plays a major role in reprogramming a normal adipocyte phenotype to inflamed adipocytes, resulting in adiposopathy [22,23,24]. Affected adipocytes proceed to terminal differentiation and inflammation, resulting in adipocyte dysfunction [23,25]. Obese animal models have low levels of HO-1 and are therefore more susceptible to oxidative stress and cytotoxicity caused by a build-up of heme, a pro-oxidant, and ROS [1,24,26].

HO-1 is the inducible isoform (HO-2 is the constitutive form) which plays a central role in maintaining beige-like adipose tissue and improves cardiovascular and liver function [27,28]. Hence, pharmacological agents that upregulate HO-1 expression have been used as a therapeutic approach for the treatment of obesity and its associated metabolic and cardiovascular diseases [27,29]. However, these interventions are not specific to adipose tissue in prevention of the inflammatory adipokines. As we and others have previously shown, lentiviruses vectors are capable of persistent transgene expression for one year in treatment of hypertension and obesity [30]. The aim of the present studies was to selectively target adipose tissue, specifically utilizing an adiponectin promoter with lentivirus for the delivery of HO-1 in growing adipose tissue, in order to reprogram the white fat to beige-like fat and decrease inflammatory adipokines. The hypothesis was that expression of HO-1, specifically to the adipocyte, would have a beneficial impact on obesity and obesity-related pathologies. Here, we demonstrated for the first time that the injection of a lentiviral–adiponectin promoter encoding HO-1 resulted in a long-lasting effect, which prevented the obesity-mediated increase of inflammation, fibrosis, and insulin resistance in HFD-fed mice. These results underscore the potential of antioxidant (HO-1) gene therapy in the treatment of obesity-derived metabolic diseases.

## 2. Materials and Methods

### 2.1. Animal Experimentation and Generation of Lentiviral Vector-Mediated HO-1 Overexpressing Mice

All animal experiments followed a protocol approved by the New York Medical College (IACUC) institutionally approved protocol, in accordance with NIH Guidelines. Six-week-old male C57BL/6J background mice, were purchased from Jackson Labs (Bar Harbor, ME, USA). For experiments, animals were divided into 3 groups (6 mice per group): lean, HFD, and HFD + Lnv-adipo-HO-1. Lean mice were fed a normal chow diet, while C57 mice were fed an HFD for 23 weeks, as previously described [31,32]. Lentiviral vectors under adiponectin-specific promoter expressing either human HO-1 or GFP adipocyte adiponectin-specific promoter were constructed by using the LentiMax^TM^ system (Lentigen, Baltimore, MA, USA and Vector Builder, Shenandoah, TX 77384, USA). The adipose tissues were targeted by 2 bolus injections into the retro orbital vein of lentiviral constructs expressing HO-1, under the control of the adiponectin promoter (Lnv-adipo-HO-1). The first injection of Lnv-adipo-HO-1 (50 µL, 1 × 10^9^ TU/mL in saline) at the 10th week and second injection into retro orbital vein (75 µL 1 × 10^9^ TU/mL in saline) at the 11th week were administered into HFD mice, while untreated HFD mice were similarly injected with Lnv-adipo-GFP vector injection control fat. Lean animals (group 1) were injected with mock virus (placebo), as previously described [33].

### 2.2. Generation of Lentiviral Vector-Mediated HO-1 Overexpressed and Deficient Adipocyte Cells for In Vitro Study

Adipo-ORF HO-1 lentivirus and Adipo-sh HO-1 (Vector builder, Shenandoah, TX 77384, USA) were applied to mice-derived adipocyte (3T3-L1) cells, as previously described [33,34] for adipocyte (3T3-L1). One million cells were seeded in 6-well plates in transducer medium that contained 1 × 10^6^ transducing units (TU) of lentiviral particles, for 3 h, to maximize the contact of lentiviral particles. Control 3T3-L1 cells were treated with the transduction medium. Cells that overexpression HO-1 or HO-1 deficient treated with 10 μg/mL puromycin. Adipo-ORF-HO-1 lentivirus and adipo-Sh-HO-1 clone were cultured. A similar procedure was used to isolate TWIST silencing, as previously described [10,35,36].

### 2.3. Measurement of Mitochondrial Oxygen-Consumption Rate in Adipocytes

The oxygen consumption rate (OCR) was measured by using an extracellular flux analyzer XFp (Seahorse Bioscience, Houston, TX, USA). Mice-derived 3T3-L1 adipocyte cells were plated at 4 × 10^5^ cells/well in a Seahorse 8-well microplate in DMEM growth media [12,34]. The medium was removed and incubated in XF assay, and after 32 min of measuring the basal respiration, oligomycin (2.5 μM) was injected. After 50 min, cells were treated with the uncoupling carboxyl-cyanide p-trifluoromethoxy phenyl hydrazine, into each well (FCCP) (1 μM), and at 74 min, rotenone (2.5 μM) was added to induce maximum respiration (uncoupled respiration), and it was followed by antimycin A, an inhibitor of ATP to decrease OCR (2.5 μM), at 98 min. ATP turnover and OCR were determined after the treatment with oligomycin and FCCP. OCR values normalized against total proteins.

PCR arrays, using the RT² Profiler™ PCR Array Mouse Adipogenesis (Qiagen, Germantown, MD USA, product no. 330231 and Cat. No. PAMM-049Z), were performed by following the manufacturers’ protocols. Gene-expression levels were compared to housekeeping as previously described using heatmap [6]. These genes are known to encoding pre-adipocyte cell markers, proliferation, differentiation and adipogenesis, lipid metabolism, and obesity [6].

Gene expression in which log10 transformed mean values lean (*n* = 3) relative to the mean values (*n* = 3) expressed in HF-mice and HF-Lnv-adipo-HO-1-lentiviral subgroups [6]. Details of this method, including Pearson correlation, are well described [6,37] at https://CRAN.R-project.org/package=gplots, and https://www.R-project.org/.

### 2.4. RNA, RT-PCR, Western Blot Analysis, Histology, and Adipocyte Cell-Size Measurements

Frozen mouse tissues, liver, kidney heart, and adipose tissue, were ground under liquid nitrogen and suspended in homogenization buffer, as previously described. Cells were lysed with lysis buffer supplemented with protease and phosphatase inhibitors. RNA, PCR, and immunoblotting for HO-1, SIRT1, MFN2, Fis1, OPA1, COX1, COX2, UCP1, TFAM, aP2, Twist1, NOV, ACC, PEG1/MEST, MnSOD, AMPK, pAMPK, AKT, and pAKT, and PGC-1α and phosphorylation of insulin receptors (IR) IRp972, IRp1146, adiponectin, β-actin, and GAPDH, were performed, as described [11,30,34,38].

### 2.5. Measurements of Oxygen Consumption, Fasting Blood Glucose, and Blood Pressure

Oxygen consumption in treated mice placed in an oxylet chamber and hourly respiratory quotients were calculated based on the VCO_2_ and VO_2_. Individual readings were recorded twice per mouse, as previously described [38]. Fasting blood glucose and blood pressure was measured by using standard tail-cuff method, as described [21,30].

### 2.6. Assessment of Vasorelaxation in Renal Interlobar Artery Rings (Myograph)

Renal interlobar arteries were cut into ring segments (2 mm in length). Vessels were contracted with phenylephrine (10^–6^ M), and vasorelaxation responses to cumulative increments in Acetylcholine (10^–8^ to 10^–4^ mol/L) concentrations were examined, as previously described [39]

### 2.7. Statistical Analysis

Data is expressed as mean ± SEM. Student’s *t*-test and one-way ANOVA with Bonferroni’s comparison were used; the null hypothesis was rejected at *p* < 0.05, and data are plotted [11,21,30].

## 3. Results

### 3.1. Lnv-adipo-HO-1 Administration Mediated Induction of HO-1 Expression Only in Adipose Tissue and Rescued HFD-Induced Phenotype and Fibrosis in Mice

Basal level of HO-1 mRNA expression in visceral adipose was similar to that of kidney and heart. After Lnv-adipo-HO-1 treatment HO-1 mRNA levels in visceral adipose tissue increased approximately three-fold, as compared to control (*p* < 0.05), while it was unchanged in the kidney and heart (Figure 1A).

Hematoxylin–eosin and Masson trichrome staining of adipose tissue of lean, HFD-fed control, and Lnv-adipo-HO-1-injected mice revealed an increase (* *p* < 0.05) in adipocyte size (hypertrophy) in HFD-fed mice as compared to lean mice (Figure 1B). Adipocyte hypertrophy was reversed by administration of Lnv-adipo-HO-1 (# *p* < 0.05 vs. control HFD-fed mice, Figure 1B). We previously described that injection of HFD-fed obese mice with Lnv-adipo-GFP does not affect adipocyte size, as compared to control obese HFD-fed mice.

### 3.2. HO-1 Mediated Co-Localization of PGC-1α: A Portion of PGC-1α Localized to the Nucleus in Adipose Tissue

For PGC-1α to act as a transcriptional co-activator it must be present in the nucleus. Figure 1C,D shows, by immunofluorescence staining (red (PGC-1α), blue (DAPI nuclei)) and arbitrary units, respectively, the subcellular localization of PGC-1α. Adipocytes from HFD-fed control mice showed reduced (* *p* < 0.05) nuclear localization of PGC-1α, as compared to lean mice. Importantly, Lnv-adipo-HO-1-transduced mice fed HFD showed a significant increase (# *p* < 0.05) in PGC-1α nuclear localization (Figure 1C,D). Moreover, both HO-1 and PGC-1α mRNA levels in adipose tissue (Figure 1E) were reduced (* *p* < 0.05) in mice fed an HFD, as compared to adipose tissues of lean mice. Increased HO-1 expression in adipose tissue of HFD-fed mice led to a normalization of adipose tissue PGC-1α mRNA levels (# *p* < 0.05 vs. untreated HFD-fed mice) (Figure 1E). As expected, the adipose tissue mRNA levels of HO-1 are significantly elevated in HO-1 overexpressing mice, as compared to both lean (* *p* < 0.05) and control HFD-fed mice (# *p* < 0.05) (Figure 1E).

### 3.3. Lnv-adipo-HO-1 Induction Decreased Weight Gain, Normalized Fasting Blood Glucose, Glucose Intolerance, Systolic Blood Pressure, and Oxygen Consumption

HFD-fed mice exhibited an average body weight gain of 2 g/week (data not shown), with an average final weight of 61.5 ± 2.4 g, significantly higher than lean mice (33.2 ± 2.1 g) (* *p* < 0.05) (Figure 1G). Lnv-adipo-HO-1 decreased the final body weight (54.1 ± 1.1 g) by an average of 7.4 g (# *p* < 0.05), as compared to untreated HFD-fed mice. Fasting blood glucose levels in lean, HFD-fed, and Lnv-adipo-HO-1-treated HFD-fed mice were 105 ± 11.8, 210 ± 14.1, and 145.5 ± 1.2 mg/dL, respectively (Figure 1I, Time 0). Lnv-adipo-HO-1 reduced the effects of an HFD on fasting blood glucose levels. Lnv-adipo-HO-1 treatment increased the tolerance to glucose challenge (*p* < 0.05) compared with control HFD-fed mice (Figure 1I). Systolic blood pressure was increased in HFD-fed mice compared to lean mice. Importantly, Lnv-adipo-HO-1 decreased blood pressure (# *p* < 0.01) in mice compared with control HFD-fed mice (Figure 1F).

We examined the effect of Lnv-adipo-HO-1 on mice fed an HFD on both O_2_ consumption and the ratio of CO_2_/O_2_. As expected, mice on an HFD displayed a significant (*p* < 0.05) decrease in VO_2_ consumption compared to lean mice. (Figure 1H). However, HFD-fed mice treated with Lnv-adipo-HO-1 exhibited a significant (*p* < 0.05) increase in oxygen consumption, with a concomitant lowering of VCO_2_/VO_2_. As seen in Figure 1J, vascular function, measured as the relaxation in response to acetylcholine, was severely impaired in renal interlobar arteries of mice fed an HFD in which group the vessels’ maximal relaxation was less than 20% at 10^−4^ M acetylcholine, as compared to vessels isolated from kidneys of lean mice with a maximal relaxation of about 80% at 10^−4^ M acetylcholine. The maximal relaxation of vessels in the Lnv-adipo-HO-1 injected mice on HFD was not significantly different from the vessels of lean mice (Figure 1J). That the Lnv-adipo-HO-1 mice restored maximal relaxation to acetylcholine to levels no different from those recorded in lean mice strengthens our finding that adipocyte-specific expression of HO-1 prevents the impairment of vascular reactivity in renal arteries of obese mice.

The effect of Lnv-adipo-HO-1 in adipose tissues extended to distal organs, including cardiac tissue. This is demonstrated by increased targeting of adipose tissue with HO-1, resulting in the restoration of cardiac HO-1, PGC1, and pAMPK, as compared to lean mice. More importantly, targeting adipose tissue with Lnv-adipo-HO-1 increased (*p* < 0.05) PGC-1α and pAMPK in obese mice (Figure 1K).

### 3.4. Lnv-adipo-HO-1 Treatment Mediated Induction of HO-1, Sirt1, PGC-1α, PRDM16, UCP1, Adiponectin, and MnSOD

Our study demonstrated reduction in HO-1 protein expression in adipose tissue of HFD-fed mice compared to control lean mice (*p* < 0.05). As seen in Figure 2, Western blot analysis demonstrated significant (*p* < 0.05) upregulation of HO-1 protein after Lenti-adipo-HO1 treatment in adipose tissue of HFD-fed mice (Figure 2). While Sirt1 and PGC-1α expression was reduced in mice adipose tissues of HFD-fed mice as compared to levels in adipose tissues of lean mice, the levels of both Sirt1 and PGC-1α were normalized in adipose tissue of HFD-fed mice and Lnv-adipo-HO-1-treated mice (Figure 2).

Uncoupling protein 1 (UCP1) was upregulated (*p* < 0.05) in adipose tissues of mice treated with Lenti-adipo-HO1 (Figure 2). We examined the effect of HFD on the brown-fat-specific gene, PRMD16, and found that PRMD16 protein expression levels were decreased (*p* < 0.05) in adipose tissue of HFD-fed control mice, an effect that was normalized in adipose tissue of Lenti-adipo-HO1 mice (Figure 2). Furthermore, Lnv-adipo-HO-1 treated mice expressed higher (*p* < 0.05) levels of MnSOD and adiponectin protein expression compared with HFD-fed mice (Figure 2).

### 3.5. Effect of Lnv-adipo-HO-1 Mitochondrial Mfn2, OPA-1, FIS-1, COX1, and COX2 Levels in Adipose Tissue of High-Fat-Diet-Fed Mice

The levels of the mitochondrial fusion-associated proteins MFN2 and OPA1, in visceral adipose tissue of mice fed an HFD were significantly (*p* < 0.05) reduced as compared to lean mice fed regular chow (Figure 3). The levels of these proteins were normalized in HFD mice treated with Lnv-adipo-HO-1, an effect that was dependent on HO-1. Conversely, Western blot results demonstrated increased levels of mito-fission related Fis1 in adipose tissue of control HFD-fed mice (Figure 3) an effect of obesity that was completely prevented in adipose tissue of the Lnv-adipo-HO-1 mice fed an HFD (Figure 3). Moreover, expression of COX-I and II proteins was decreased in HFD-fed mice (*p* < 0.05), an effect that was reversed in Lnv-adipo-HO-1-treated mice adipose tissue (Figure 3).

### 3.6. Lnv-adipo-HO-1 Administration on Expression of Adipogenic and Inflammatory Mediators in Adipose Tissue of Obese Mice

Aberrant expression of nephroblastoma overexpressed (NOV) and Twist1 are evident in inflammation and obesity. As seen in Figure 4, Twist1 and NOV protein expression in adipose tissue of Lnv-adipo-HO1-treated mice was reduced (*p* < 0.05) as compared with HFD-fed mice adipose tissue (Figure 4). Moreover, Western blot analysis demonstrated increased expression of adipocyte protein 2 (ap2) and mesoderm-specific transcript (Mest) protein expression in adipose tissue of HFD-fed mice, as compared to levels in adipose tissues of lean mice. HFD-fed mice treated with Lnv-adipo-HO-1 had significantly (*p* < 0.05) diminished protein expression of both MEST and ap2 in adipose tissue compared with control HFD-fed mice (Figure 4). As shown in Figure 5, mRNA levels of TNF-α, IL1β, CCL2, and NOV in HFD-fed mice were increased (*p* < 0.05) compared with lean mice. The expression of these genes was reduced (*p* < 0.05) in adipose tissues of Lnv-adipo-HO1-HFD-fed mice. Moreover, HFD-fed mice displayed an increase (*p* < 0.05) in adipose tissue fibrosis compared to lean mice, an effect of HFD that was reduced (*p* < 0.05) in Lnv-adipo-HO-1-injected HFD-fed mice (Figure 5).

### 3.7. Effect of Lnv-adipo-HO-1 Administration on pACC, pAKT, pAMPK, and Insulin Receptor Phosphorylation

An HFD decreased phosphorylation of ACC, AKT, and AMPK protein levels, compared with control lean mice (*p* < 0.05; Figure 6). Transduction by Lnv-adipo-HO-1 normalized the phosphorylation of pACC, pAKT, and pAMPK (Figure 6). Adipose tissue of HFD-fed mice exhibited lower protein levels pIR972 and pIR1146, which was normalized by β-actin. Lnv-adipo-HO-1 induction produced a significant (*p* < 0.05) increase in protein expression of pIR972 and pIR1146 (Figure 6).

### 3.8. HO-1 Regulation of Mitochondrial Biogenesis, cyp2C44, and Twist1 in Adipocyte Cell Culture

To further elucidate the central role of HO-1 on mitochondrial function in adipocyte cells, we modified HO-1 levels by both lentivirus-mediated HO-1 downregulation and upregulation in pre- adipocyte cells in vitro. Whereas knockdown of HO-1 caused a reduction (*p* < 0.05) in the levels of CYP2C44, PGC-1α, MFN1, and MFN2, overexpression of HO-1 caused an increase (*p* < 0.05) in the mRNA levels of these genes, beyond the levels observed in control cells (Figure 7A–F). Conversely, mRNA levels of Fis1 and TWIST1 were increased (*p* < 0.05) in HO-1-deficient cells and reduced (*p* < 0.05) in cells overexpressing HO-1 (Figure 7A–F).

### 3.9. HO-1 Regulation of Mitochondrial Function in Adipocyte Cell Culture

To ascertain if the HO-1 KD adipocyte cells show defective mitochondrial bioenergetics, we measured mitochondrial respiration in WT, HO-1 KD, and adipocyte cells overexpressed HO-1. Real-time oxygen-consumption rates (OCRs) in adipocyte cells show that basal respiration, representing the sum of all physiological mitochondrial oxygen consumption, was decreased in the mitochondria from HO-1 deficient cells, indicating lower respiratory function compared with WT cells, which was rescued after overexpression of HO-1 (Figure 7G). The injection of oligomycin led to a decrease in OCR. FCCP was used to generate ATP (Figure 7G) uncouples respiration from oxidative phosphorylation used to measure maximal OCR. HO-1 KD cells showed a lower basal level OCR (Figure 7H), which was rescued after overexpression of HO-1 (Figure 7G,I), that displays reduction in the mitochondrial activity. Non-mitochondrial OCR was determined by inhibiting the respiratory chain, using rotenone and antimycin A. ATP turnover was significantly decreased in HO-1-ablated cells (Figure 7I). The maximum respiration was also significantly (*p* < 0.05) lower in HO-1-deficient cells which were rescued after overexpression of HO-1. Mitochondria isolated from HO-1-ablated cells displayed a reduction in coupling efficiency that drives ATP synthesis, which was rescued after overexpression of HO-1 (Figure 7) (*p* < 0.05).

### 3.10. TWIST1 Expression Regulated HO-1 and PGC-1α in Cultured Adipocyte Cells

As HO-1 regulated TWIST1 expression levels, we examined whether TWIST1 expression could impact the expression of HO-1 and its upstream regulating genes, CYP2C44 and PGC-1α. As seen in Figure 8, knockdown of TWIST1 in adipocytes (Figure 8A) caused an upregulation (*p* < 0.05) in the mRNA levels of CYP2C44, PGC-1α and HO-1 as compared to control cells (Figure 8B–D). Moreover, the real-time oxygen consumption rate in TWIST1-deficient adipocyte cells was elevated, especially the maximal oxygen consumption-rate (after the uncoupling of respiration from oxidative phosphorylation), as compared to control cells (Figure 8E).

### 3.11. RNA Array Analyses Identify Changes in Correlation Coefficients of Gene Expression in Lean (C), High-Fat (HF), and Heme Oxygenase-1 (HO-1) Lenti-Virus Groups

Our study investigated the role of HO-1 during the adipogenic process and analyzed the expressions of 88 genes that expressed only in adipocytes before and during adipogenesis, as detailed in [6]. The results for the Lnv-adipo-HO-1-injected HFD-fed mice demonstrate marked attenuation of the changes observed in HFD-fed control mice (Figure 9 and Figure 10). Very clearly, the top 10 upregulated genes correlated strongly positively with each other, and the top 10 downregulated genes correlated positively with each other, but very negatively with the other group (Figure 9 and Figure 10).

### 3.12. The mRNA Expression Levels as a Result of HO-1 Upregulation in Adipose Tissues of Lean, HFD-Fed, and Lnv-adipo-HO-1-HFD-Fed Mice

As seen in Figure 11, the mRNA levels of *Sfrp5*, *wnt5b*, *Nr1h3*, *Twist1*, *Wnt10b*, *Lmna*, *Rb1*, *Rxra*, and *Jun* were upregulated in adipose tissues of HFD-fed mice, Group 2, as compared to the levels in adipose tissues of lean mice, Group 1 (* *p* < 0.05 and ** *p* < 0.005), and were significantly reduced (* *p* < 0.05 and ** *p* < 0.005) in adipose tissues of Lnv-adipo-HO-1-HFD-fed mice as compared to HFD-fed mice, Group 3. Moreover, the mRNA expression levels of some genes, i.e., *Sirt1*, *Slc2a4*, *Ppard*, *Adipoq*, *UCP1*, *Shh*, *PRDM16*, *Lpl*, *Insr*, and *Taz* were downregulated (* *p* < 0.05 and ** *p* < 0.005) in adipose tissues of HFD-fed mice, Group 2, as compared to the levels in adipose tissues of lean mice, Group 1. As expected, these genes were upregulated (* *p* < 0.05 and ** *p* < 0.005) in adipose tissues of Lnv-adipo-HO-1-HFD-fed mice, Group 3, as compared to the mRNA levels in adipose tissues of HFD-fed mice, Group 2 (Figure 11).

## 4. Discussion

In the present study, we demonstrate that targeting adipocyte-specific HO-1 gene expression in mice fed an HFD displayed a pronounced metabolic improvement, in spite of food intake being unaltered between the groups. This was manifested by a reduction in body weight and fasting blood glucose levels, and increased glucose tolerance and oxygen consumption. Selective gene targeting of the adipocyte by HO-1 was associated with decreased adipocyte hypertrophy, as measured by a 40% decrease in adipocyte cell size compared to mice fed an HFD. The Lnv-adipo-HO-1 expression-reduced hypertrophy was associated with decreased levels of the inflammatory molecules, NOV, TWIST, and TNFα. Targeted Lnv-adipo-HO-1 expression correlated positively with an improvement in the vasodilatory response to acetylcholine, increased insulin sensitivity, and increased heart mitochondrial proteins, suggesting that selective expression of HO-1 in adipocytes has a positive impact in distal organs. An increase of beige-like adipocyte-associated marker genes, energy expenditure, adipocyte respiration in vivo and in adipocyte cell culture, and mitochondrial biogenesis was measured by COX-1 and COX-2. In contrast, knockdown of HO-1 in adipocytes resulted in decreased mitochondrial biogenesis, reduced mitochondrial oxygen consumption, and CYP2C44 levels, but increased levels of pro-inflammatory TWIST1. Finally, RNA microarray and heatmap analysis of adipocyte gene-specific target genes indicate the presence of a protein network that is under control of HO-1 and is depressed in obesity-mediated metabolic dysfunction, which is reversed by the specific targeting of HO-1 in adipocytes.

In this study, we demonstrate that adipocyte-specific HO-1 expression improves adipocyte hypertrophy by decreasing cell size (Figure 1); the progression of obesity is tightly associated with increased adipocyte hypertrophy [24,25,40]. An increase in adipocyte cell size is negatively correlated with adiponectin levels [22]. Further, a reduction of adipocyte hypertrophy is a measure of reprogramming of the white adipocyte phenotype to a beige-like phenotype, which expresses increased levels of mitochondrial, thermogenic, and biogenic genes involved in the stimulation of ATP production and mitochondrial respiration (Figure 9). UCP1 is crucial in maintaining thermogenic pathways that are essential for adipocyte function [41,42,43]. An increase in HO-1 expression increased expression of carnitine transporter and the ATP/ADP ratio [44]. A decrease in adipocyte carnitine transporter levels impairs thermogenesis and mitochondrial function and increases obesity [45,46]. The beiging of white fat is related to decreases in adipose expansion and increased insulin sensitivity that involves increased expression of PRMD16 [47] and of CYP2C44-gene [22,48].

Our findings indicate that the increased expression of HO-1 in adipocytes leads to insulin sensitivity. This is supported by the augmented phosphorylation status of insulin-receptor-associated signaling molecules (ACC, AKT, AMPK, IR p972, and IR p 1146). These data support the concept that adipose tissue releases molecules that benefit distal organs, including the vascular system.

Obesity alone is capable of reducing the HO-1 expression and activity, resulting in insulin resistance in both humans and mice [6,19]. As noted above, the direct consequence of the upregulation of HO-1 in adipocytes is an increase in the translocation of PGC-1α into the adipocyte nucleus, leading to an increased expression of nuclear transcription factors, including mitochondrial genes. PGC-1α acts as a dominant regulator of mitochondrial biogenesis and thermogenesis by inducing the expression of UCP1 and key enzymes of the mitochondrial respiratory chain [49].

In the current report, specifically targeting adipocytes by HO-1 is associated with a decrease of NOV, a multifunctional protein involved in inflammation and interstitial fibrosis. High levels of NOV have been attributed to obesity, and its ablation reduced inflammation and fat mass and increased insulin sensitivity [11,38]. NOV is abundantly expressed in visceral fat, which contributes to vascular dysfunction and a failing heart [11]. Furthermore, ablation of HO-1 in adipocyte cell culture resulted in decreased levels of PGC-1α and Mfn2, while levels of Fis1 and TWIST increased significantly. Adipocyte-specific HO-1 increased expression resulted in marked reduction in TWIST and Fis1 levels. In addition, oxygen consumption rates (OCR) were increased by ablation of TWIST (Figure 8).

Additionally, we report that CYP2C44 levels were markedly decreased by HO-1 ablation, an effect that was reversed and normalized by increased adipocyte-specific levels of HO-1. This is of significance because the reduced expression of CYP2C44 leads to the reduced conversion of arachidonic acid into EETs, a detrimental blow to mitochondrial biogenesis, as EETs promote mitochondrial biogenesis [50].

RNA arrays (88 genes) in adipocytes of visceral fat isolated from HFD-fed Lnv-adipo-HO-1 mice showed a pattern of gene expression that elucidated the mechanisms by which adipocyte HO-1 prevents HFD-driven metabolic dysfunction. Results from Figure 11 and Figure 12 indicated that HO-1 reversed HFD-driven downregulation of SIRT1, LPL, Adiponectin, UCP1, and PRDM16. Thus, our study shows that a set of new genes and biomarkers are an emerging target for the treatment of obesity-mediated metabolic diseases.

## 5. Conclusions

In conclusion, adipocyte-specific expression of HO-1 appears to be a master regulator of adipocyte phenotype and, in turn, is capable of inducing systemic changes in vascular and insulin sensitivity through the action of adipocyte resident signaling pathways (Figure 12). Our results clearly demonstrate that the specific expression of HO-1 in adipocytes has a major impact on distal organs Figure 12. A constellation of findings of HO-1 gene therapy in the obese-mice model clearly demonstrates that targeting adipose tissue with HO-1 gene can drastically influence adipocyte phenotype and reprogram visceral adipocytes to healthy beige-like cells that are capable of anti-inflammatory effects, presumably by promoting the adipocyte release of bioactive molecules that, in turn, impact distal organ function beneficial to reversing the course of metabolic diseases; this process still remains to be elucidated. Lentiviral vectors achieve durable expression of transgene, i.e., HO-1, the increase antioxidants, and reduction of inflammatory adipokines, may have potential for clinical application in humans with HO-1 deficiency.

## Figures and Tables

**Figure 1 antioxidants-09-00040-f001:**
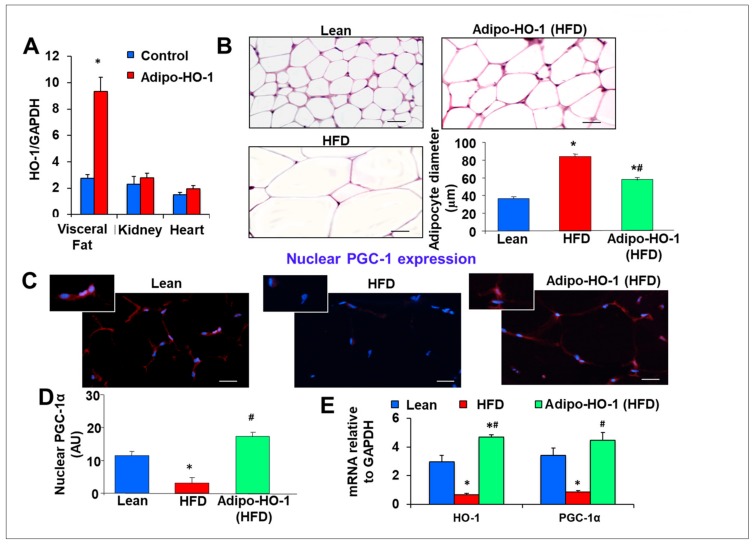
Effects of adipocyte-specific overexpression of HO-1 (heme oxygenase-1) in high-fat diet (HFD)-fed mice. (**A**) HO-1 mRNA expression levels in visceral fat * *p* < 0.05 vs. control, kidney and heart in Lnv-adipo-HO-1, *n* = 5. Comparisons of adipose tissues of lean, HFD-fed, and Lnv-adipo-HO-1 (HFD)-fed mice: (**B**) histological analysis of adipocyte diameter, using hematoxylin-eosin and Masson-trichrome staining; (**C**) immunofluorescence photomicrographs of PGC-1α expression (red staining), scale bar 20 μm (10×). (**D**) Immunomorphometrical measurement of nuclear localization of PGC-1 (AU); (**E**) mRNA levels of HO-1 and PGC-1α. (**F**) Body weight (BW); (**G**) fasting glucose (at 10) and glucose tolerance test; **(H**) Systolic blood pressure; (**I**) oxygen consumption (VO2); (**J**) acetylcholine-mediated relaxation of renal interlobar arteries *n* = 5, * *p* < 0.05 vs. lean, # *p* < 0.05 versus fat. (**K**) Representative Western blot analysis of PGC-1α, HO-1, pAMPK, and AMPK in heart-tissue lysates of lean, HFD-fed, and Lnv-adipo-HO-1 HFD-fed mice; *n* = 5, * *p* < 0.05 versus lean; # *p* < 0.05 versus HFD alone.

**Figure 2 antioxidants-09-00040-f002:**
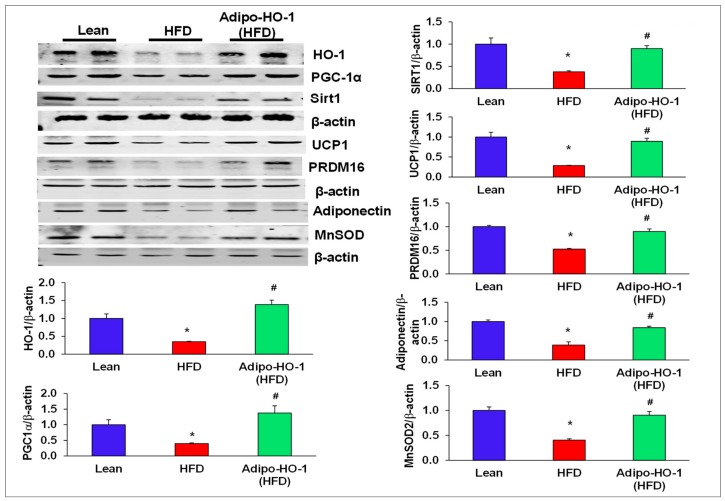
Lnv-adipo-HO-1 treatment mediated induction of key proteins involved in adipocyte metabolism. Representative Western blot analysis of HO-1, PGC-1α, Sirt1, MnSOD, PRDM16, UCP-1, and MnSOD with their corresponding β-actin in adipose tissue of lean, HFD, and Lnv-adipo-HO-1 (HFD) mice; *n* = 5, * *p* < 0.05 versus lean; # *p* < 0.05 versus HFD alone.

**Figure 3 antioxidants-09-00040-f003:**
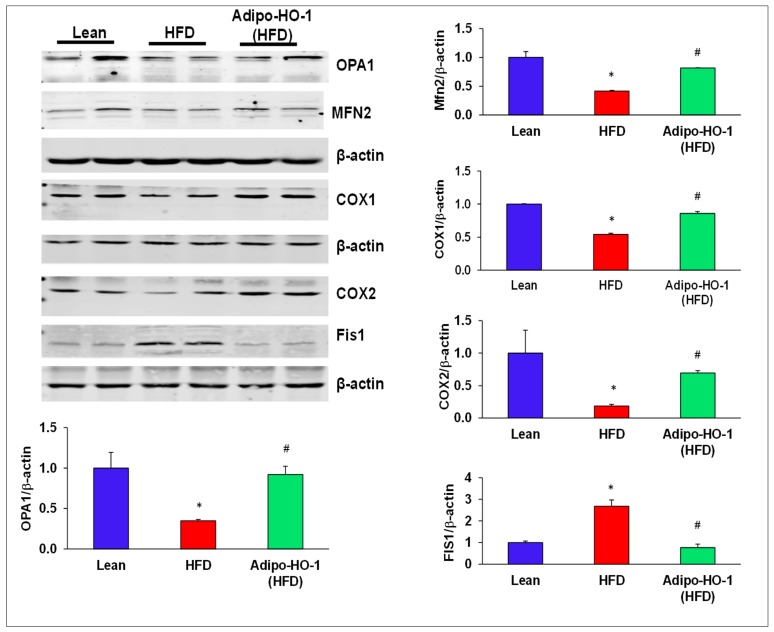
Effect of Lnv-adipo-HO-1 of key mitochondrial protein levels in adipose tissues of Lnv-adipo-HO-1 (HFD) mice. Representative Western blot analysis of OPA1, MFN2, COX1, COX2, and Fis1 in adipose tissue of lean, HF, and Lnv-adipo-HO-1 (HFD) mice; *n* = 5, * *p* < 0.05 versus lean; # *p* < 0.05 versus HFD alone.

**Figure 4 antioxidants-09-00040-f004:**
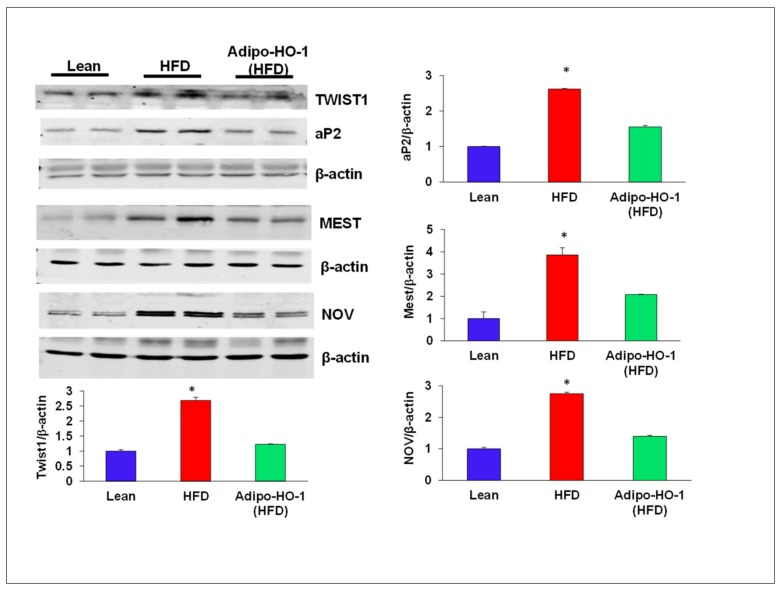
Expression of adipogenic and inflammatory mediators in adipose tissue of lean, HFD, and Lnv-adipo-HO-1 (HFD) mice. Representative Western blot analysis of TWIST1, aP2, MEST, and NOV and their respective β-actin in adipose tissue of lean, HFD, and Lnv-adipo-HO-1 (HFD) mice; *n* = 5, * *p* < 0.05 versus WT and Lnv-adipo-HO-1 (HFD).

**Figure 5 antioxidants-09-00040-f005:**
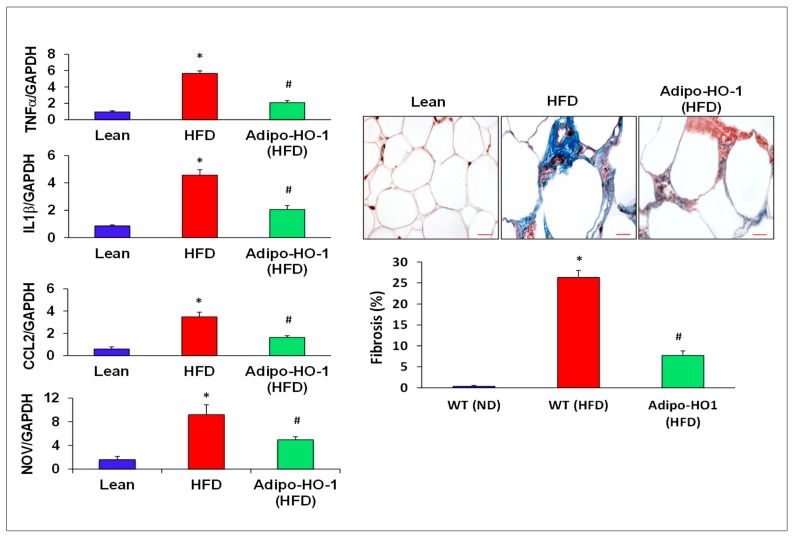
Expression of inflammatory mediators in adipose tissue of lean, HFD, and Lnv-adipo-HO-1 (HFD) mice. Relative mRNA levels of TNF-α, IL1β, CCL2, and NOV, followed by histological analysis of adipose tissue fibrosis, using hematoxylin/eosin and Masson trichrome staining; *n* = 5, * *p* < 0.05 versus lean; # *p* < 0.05 versus HFD alone.

**Figure 6 antioxidants-09-00040-f006:**
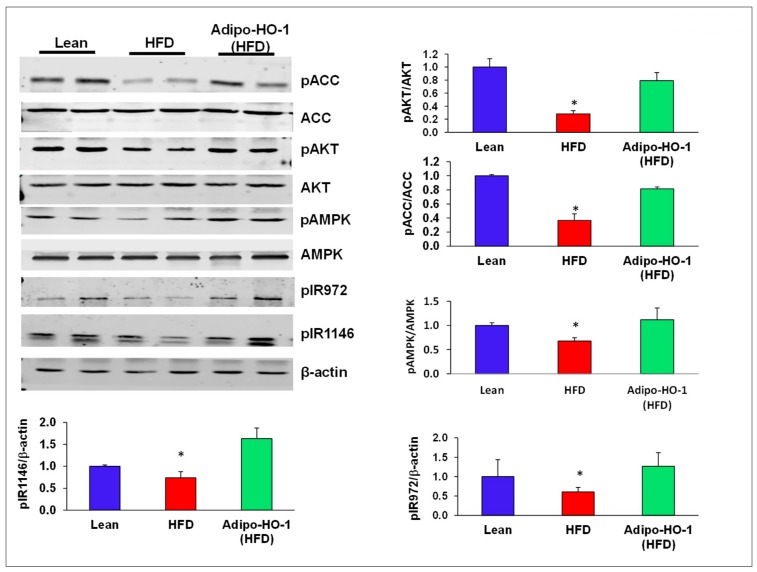
Representative Western blot analysis of insulin receptor phosphorylation and key phosphorylated proteins within the insulin receptor signaling cascade in adipose tissue of lean, HFD, and Lnv-adipo-HO-1 (HFD) mice. Representative Western blots and fluorometric analysis relative to β-actin of pACC, ACC, pAKT, AKT, pAMPK, AMPK, pIR972, and pIR1146; *n* = 5, * *p* < 0.05 versus lean and Lnv-adipo-HO-1 (HFD) mice.

**Figure 7 antioxidants-09-00040-f007:**
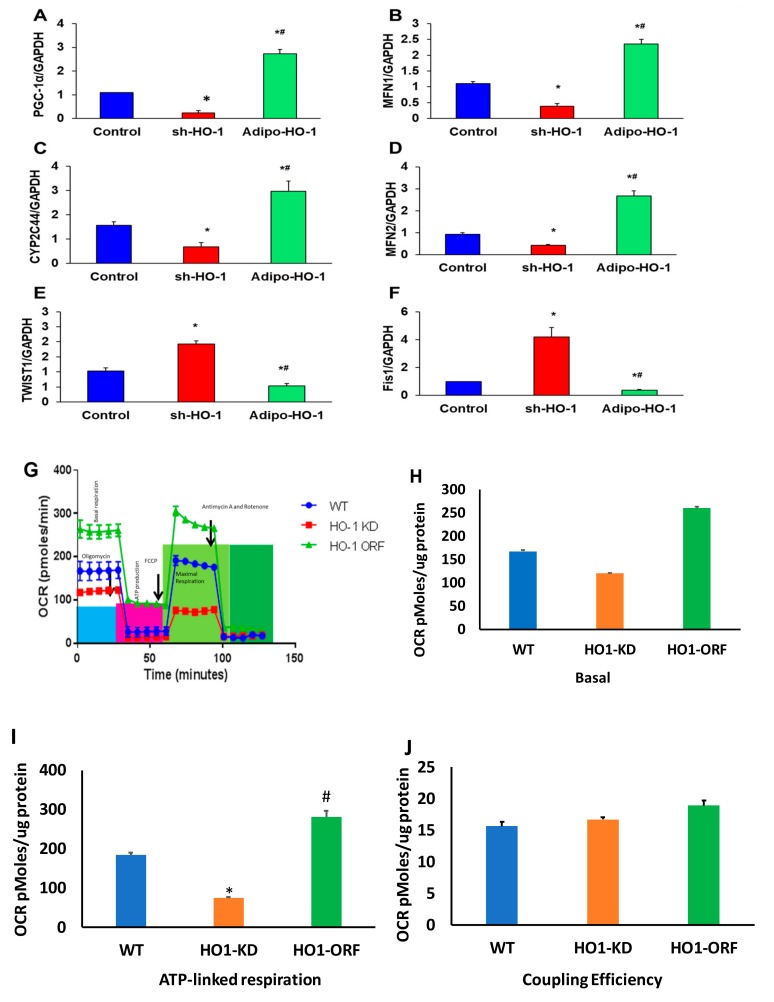
Heme oxygenase-dependent regulation of mitochondrial dynamics and inflammation in vitro. Quantitative gene expression analysis of (**A**) PGC-1α, (**B**) MFN1, (**C**) Cyp2c44, (**D**) MFN2, (**E**) Twist1, and (**F**) Fis1, *n* = 3, * *p* < 0.05 versus control; # *p* < 0.05 versus sh-HO-1. (**G**) HO-1 regulation of mitochondrial function in adipocyte cell culture. (**H**) Basal of OCR, (**I**) ATP-linked respiration, and (**J**) coupling efficiency. Oxygen-consumption rates (OCR) of sh-HO-1 (HO-1 KD), Lnv-adipo-HO-1 (HO-1 ORF), and control cultured adipocytes, *n* = 3, * *p* < 0.05 versus control, # *p* < 0.05 vs. HO-1 KD.

**Figure 8 antioxidants-09-00040-f008:**
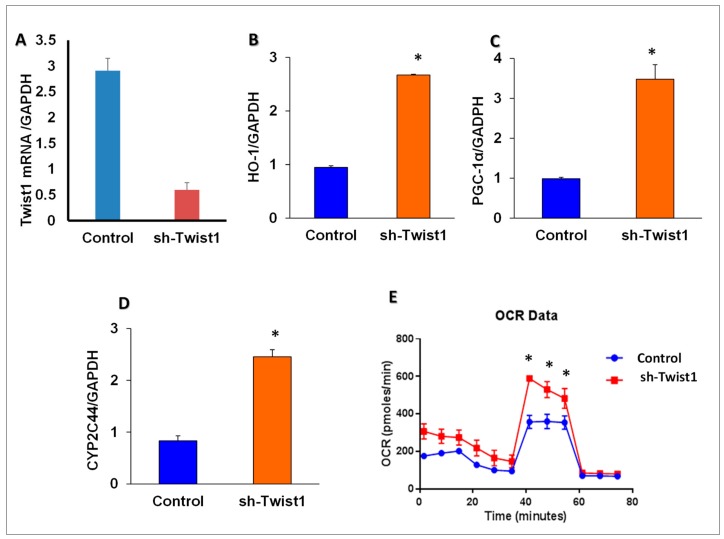
(**A**) TWIST silencing, effect of TWIST1 silencing on HO-1, PGC-1α, CYP2C44 in cultured adipocyte cells. mRNA expression of (**B**) HO-1, PGC-1α, (**C**) and (**D**) CYP2C44, after shRNA-mediated knockdown of Twist1 and (**E**) oxygen consumption rates (OCR) of sh-TWIST1 and control cultured adipocytes, *n* = 3, * *p* < 0.05 versus control.

**Figure 9 antioxidants-09-00040-f009:**
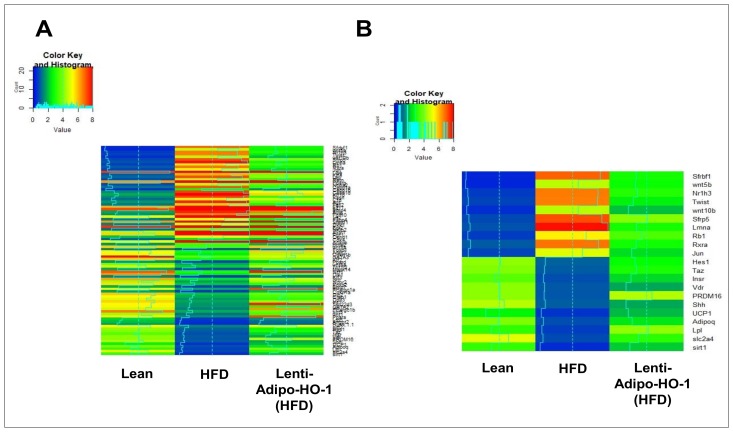
RNA-array-analyses changes in correlation coefficients gene expression in lean, HFD, and Lnv-adipo-HO-1 (HFD) groups. (**A**) RNA-array-analyses changes in correlation coefficients gene expression of 86 genes involved in Fat Metabolism. (**B**) RNA array analysis showing changes in correlation coefficients for gene expression of sfrbf1, Wnt5b, Nr1h3, Twist, Wnt10b, Sfrp5, Lmna, Rb1, Rxra, Jun, Hes1, Taz, Insr, Vdr, PRDM16, Shh, UCP1, Adipoq, Lpl, Slc2a4, and Sirt1 in mice injected with Lnv-adipo- HO-1.

**Figure 10 antioxidants-09-00040-f010:**
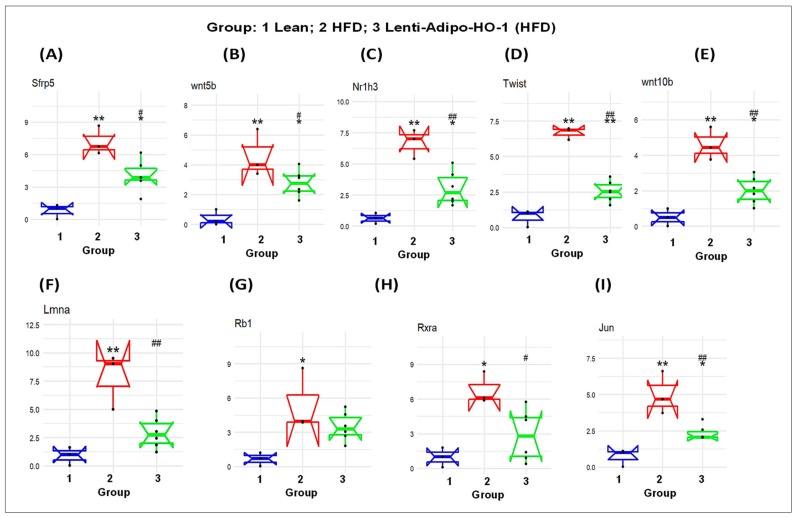
The mRNA expression of lean (Group 1) mice, HFD (Group 2) mice, and Lnv-adipo-HO-1 (HFD) mice (Group 3). (**A**) sfrp5, (**B**) wnt5b, (**C**) Nr1h3, (**D**) Twist, (**E**) Wnt10b. (**F**) Lmna (**G**) Rb1 (**H**) Rxra (**I**) Jun in adipose tissue. (*n* = 5, * *p* < 0.05, ** *p* < 0.005 vs. lean; # *p* < 0.05, ## *p* < 0.005 vs. HFD alone).

**Figure 11 antioxidants-09-00040-f011:**
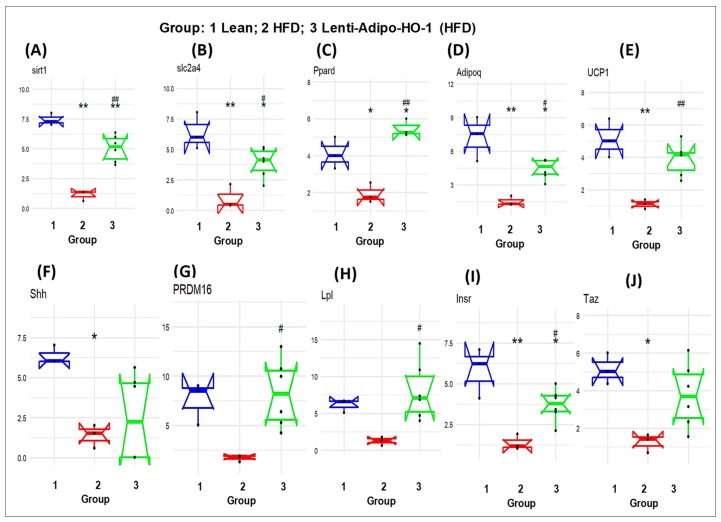
The mRNA expression of (Group 1) mice, HFD (Group 2) mice, and Lnv-adipo-HO-1 (HFD) mice (Group 3). (**A**) sirt1, (**B**) slc2a4, (**C**) Ppard, (**D**) Adipoq, (**E**) UCP1, (**F**) Shh, (**G**) PRDM16, (**H**) Lpl, (**I**) Insr, and (**J**) Taz. (*n* = 5, * *p* < 0.05, ** *p* < 0.005 vs. lean; # *p* < 0.05, ## *p* < 0.005 vs. HFD alone).

**Figure 12 antioxidants-09-00040-f012:**
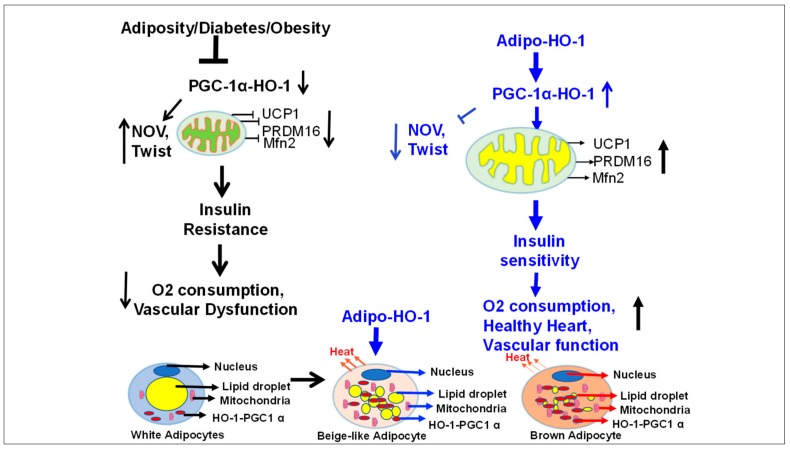
Schematic presentation of postulated mechanisms by which adipocyte-specific HO-1 expression promotes the reprogramming of adipose tissue obese phenotype to a healthy brown-fat phenotype. Obesity leads to a marked suppression of HO-1 within adipocytes, resulting with subsequent increases of inflammatory adipokines, including NOV and TWIST, and leading to suppression of thermogenic genes and mitochondrial dysfunction. These changes at the level of adipocytes are accompanied with reduced energy consumption, impaired vascular function, and insulin resistance. A targeted overexpression of HO-1 within adipocytes prevents and reverses these changes by stimulating PGC1-α expression, leading to an increase in thermogenic genes (UCP1 and PRDM16) and a decrease in inflammatory adipokines, (NOV and TWIST). This, in turn, increases mitochondrial respiration, oxygen consumption, and insulin receptor phosphorylation, all of which contribute to improving vascular function and increasing expression of heart protective genes. Hence, selective expression of adipocyte HO-1 offers a multifactorial clinical approach to the treatment of obesity concomitant metabolic disorders.

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
