# Peer review of "Adipocyte Specific HO-1 Gene Therapy Is Effective in Antioxidant Treatment of Insulin Resistance and Vascular Function in an Obese Mice Model"

_antioxidants, 2020, doi:10.3390/antiox9010040_

Round 1

Reviewer 1 Report

The manuscript entitled “Adipocyte Specific HMOX1 Gene Therapy is Effective in Antioxidant Treatment of Insulin Resistance and Vascular Function in an Obese Mice Model” examined effects of adipocyte specific heme oxigenase-1 expression against high fat diet induced obesity. The heme oxigenase-1 overexpression improved the phenotypes of the high fat diet induced obesity mice. These results suggest that the heme oxigenase-1 expression induced by chemicals or transgenic techniques contributes to treatment of patients with obesity.

The study includes important information for benefits of the heme oxigenase-1 expressed in adipocyte against obesity.

Comments:

1). Abbreviation of heme oxygenase-1 should be unified in HO-1 or HMOX-1.

2). The authors should improve the figures for more clarity. The x- and y-axis should be described.

Author Response

Reviewer 1

Comments and Suggestions for Authors

The manuscript entitled “Adipocyte Specific HMOX1 Gene Therapy is Effective in Antioxidant Treatment of Insulin Resistance and Vascular Function in an Obese Mice Model” examined effects of adipocyte specific heme oxigenase-1 expression against high fat diet induced obesity. The heme oxigenase-1 overexpression improved the phenotypes of the high fat diet induced obesity mice. These results suggest that the heme oxigenase-1 expression induced by chemicals or transgenic techniques contributes to treatment of patients with obesity.

The study includes important information for benefits of the heme oxigenase-1 expressed in adipocyte against obesity.

Comments:

1). Abbreviation of heme oxygenase-1 should be unified in HO-1 or HMOX-1.

 This has been corrected to HO-1 throughout the manuscript.

2). The authors should improve the figures for more clarity. The x- and y-axis should be described.

 Figures 10 and 11 are now darker and clearer.

Submission Date

06 November 2019

Date of this review

18 Nov 2019 17:01:54

Reviewer 2 Report

Dear editor,

about the manuscript entitled “Adipocyte Specific HMOX1 Gene Therapy is Effective in Antioxidant Treatment of Insulin Resistance and Vascular Function in an Obese Mice Mode”, by Shailendra P. Singh and coworkers, enclosed my suggestions:

INTRODUCTION: “Obesity has become highly prevalent in the past decade worldwide, affecting all age groups and populations”. PLEASE USID A VALID REFERENCE TO SUPPORT YOUR SENTENCE. Please add a definition of obesity. As first, in obese patients there is a local and systemic over expression of inflammatory proteins and anti-apoptotic proteins (as sirtuins) that might condition the clinical outcomes also in patients without heart disease and with normal cardiac performance (Front Physiol. 2018 Aug 21;9:1030. doi: 10.3389/fphys.2018.01030; Int J Mol Sci. 2019 Mar 6;20(5). pii: E1153. doi: 10.3390/ijms20051153). This might result in reduction of cardiac performance and worse prognosis in patients with altered glycemic control and insulin resistance (Front Physiol. 2018 Aug 21;9:1030. doi: 10.3389/fphys.2018.01030; Int J Mol Sci. 2019 Mar 6;20(5). pii: E1153. doi: 10.3390/ijms20051153). However, specific therapies to target this over-inflammation at level of abdominal adipose tissue might be valid to improve clinical outcomes in affected patients (Front Physiol. 2018 Aug 21;9:1030. doi: 10.3389/fphys.2018.01030; Int J Mol Sci. 2019 Mar 6;20(5). pii: E1153. doi: 10.3390/ijms20051153). Please discuss this point. In addition, it is relevant to remember the importance of the epericardial fat surrounding coronary arteries, as source of inflammatory molecules and as determinant of worse prognosis of patients with insulin resistance and treated by coronary arteries revascularization (Cardiovasc Diabetol. 2019 Sep 30;18(1):126. doi: 10.1186/s12933-019-0931-0). Intriguingly, also in this pathogenic condition the hypoglycemic drug therapy might be reduce and control the peri-coronary flat inflammation and it might ameliorate clinical outcomes (Cardiovasc Diabetol. 2019 Sep 30;18(1):126. doi: 10.1186/s12933-019-0931-0). Please discuss this point. Intriguingly, the sirtuin might directly affect the inflammatory activity and plaque functionality in diabetic atherosclerotic plaques (Diabetes. 2015 Apr;64(4):1395-406. doi: 10.2337/db14-1149), and the thrombus burden in patients with diabetes and acute coronary syndrome (J Cell Physiol. 2019 Jul 11. doi: 10.1002/jcp.29064). Please add these informations in the text. Notably, a relevant link exists between markers of inflammation, metabolic dysfunction of adipose tissue and insuin resistance, as in the case of new PCI in subjects with normal glucose tolerance, in those the adiponectin and insulin resistance are related to restenosis (Cardiovasc Diabetol. 2019 Mar 4;18(1):24. doi: 10.1186/s12933-019-0826-0). This is indicative of a tight correlation between adipose tissue dysfunction, inflammation, insulin resistance and worse prognosis also in absence of altered glucose homeostasis and diabetes. please discuss this point. What is your observation? The INTRODUCTION is too long. Please short it at 1 and half page. STATISTICAL ANALYSIS: about the sentence “Data is expressed as mean ± S.E.M. Student’s t-test was used for pairwise..” you have to use the verbs in past and/or all in the present. Please correct it. RESULTS: in the figure 1, please indicate how much is the scale used and the zoom (x10? X 20?). it is not clear. Report it in the images. The results are too long, please include a shorter description of main study results, and if possible please use a re-submission of all study outcomes in a supplementary file. This might be used to deeply explained all study outcomes and results, because in this way that you used it might result more confused. Please do not include figure 12 in the Conclusions. I prefer to see it in results, and/or in the proposed Supplementary files. Please correct it. You might use a table with the most and relevant study results. It might make the difference. What are study limitations? Please discuss it.

Author Response

Reviewer 2

Comments and Suggestions for Authors

Dear editor,

about the manuscript entitled “Adipocyte Specific HMOX1 Gene Therapy is Effective in Antioxidant Treatment of Insulin Resistance and Vascular Function in an Obese Mice Mode”, by Shailendra P. Singh and coworkers, enclosed my suggestions:

INTRODUCTION: “Obesity has become highly prevalent in the past decade worldwide, affecting all age groups and populations”. ”. Included the proper citation.

 Please add a definition of obesity. In human obesity defined in manuscript as a BMP above 32 As first, in obese patients there is a local and systemic over expression of inflammatory proteins and anti-apoptotic proteins (as sirtuins) that might condition the clinical outcomes also in patients without heart disease and with normal cardiac performance Thomazo JB, Contreras Pastenes J, Pipe CJ, Le Révérend B, Wandersman E, Prevost AM. Probing in-mouth texture perception with a biomimetic tongue. J R Soc Interface. 2019;16(159):20190362. doi:10.1098/rsif.2019.0362). This might result in reduction of cardiac performance and worse prognosis in patients with altered glycemic control and insulin resistance Thomazo JB, Contreras Pastenes J, Pipe CJ, Le Révérend B, Wandersman E, Prevost AM. Probing in-mouth texture perception with a biomimetic tongue. J R Soc Interface. 2019;16(159):20190362. doi:10.1098/rsif.2019.0362).  However, specific therapies to target this over-inflammation at level of abdominal adipose tissue might be valid to improve clinical outcomes in affected patients (Front Physiol. 2018 Aug 21;9:1030. doi: 10.3389/fphys.2018.01030; Int J Mol Sci. 2019 Mar 6;20(5). pii: E1153. doi: 10.3390/ijms20051153). Please discuss this point. This point is beyond the scope of our manuscript as SIRTUINS is very interesting but our focus is on adipocyte targeting HO-1.    In addition, it is relevant to remember the importance of the epericardial fat surrounding coronary arteries, as source of inflammatory molecules and as determinant of worse prognosis of patients with insulin resistance and treated by coronary arteries revascularization (Cardiovasc Diabetol. 2019 Sep 30;18(1):126. doi: 10.1186/s12933-019-0931-0). We value your comment but it is not related to our current manuscript. Intriguingly, also in this pathogenic condition the hypoglycemic drug therapy might be reduce and control the peri-coronary flat inflammation and it might ameliorate clinical outcomes (Cardiovasc Diabetol. 2019 Sep 30;18(1):126. doi: 10.1186/s12933-019-0931-0). Please discuss this point. Intriguingly, the sirtuin might directly affect the inflammatory activity and plaque functionality in diabetic atherosclerotic plaques (Diabetes. 2015 Apr;64(4):1395-406. doi: 10.2337/db14-1149), and the thrombus burden in patients with diabetes and acute coronary syndrome (J Cell Physiol. 2019 Jul 11. doi: 10.1002/jcp.29064). Please add these informations in the text. Notably, a relevant link exists between markers of inflammation, metabolic dysfunction of adipose tissue and insuin resistance, as in the case of new PCI in subjects with normal glucose tolerance, in those the adiponectin and insulin resistance are related to restenosis (Cardiovasc Diabetol. 2019 Mar 4;18(1):24. doi: 10.1186/s12933-019-0826-0). This is indicative of a tight correlation between adipose tissue dysfunction, inflammation, insulin resistance and worse prognosis also in absence of altered glucose homeostasis and diabetes. please discuss this point. We agree with the reviewer but this is beyond our goal. What is your observation? The INTRODUCTION is too long. Please short it at 1 and half page. STATISTICAL ANALYSIS: about the sentence We shortened the introduction. “Data is expressed as mean ± S.E.M. Student’s t-test was used for pairwise..” you have to use the verbs in past and/or all in the present. This has been corrected. Please correct it. RESULTS: in the figure 1, please indicate how much is the scale used and the zoom (x10? X 20?). it is not clear. Report it in the images. This is indicated X10. The results are too long, please include a shorter description of main study results, and if possible please use a re-submission of all study outcomes in a supplementary file. This might be used to deeply explained all study outcomes and results, because in this way that you used it might result more confused. The results cannot be submitted in a supplementary file.  Please do not include figure 12 in the Conclusions. I prefer to see it in results, and/or in the proposed Supplementary files. Figure 12 is more effective to the reader and the number of citations.  I care about the citation as this reflects on the journal impact factor. Please correct it. You might use a table with the most and relevant study results. It might make the difference. What are study limitations? Please discuss it. Limitation is how to apply gene therapy for human obesity as indicated in the last statement.

Submission Date

06 November 2019

Date of this review

12 Nov 2019 08:48:25

Reviewer 3 Report

This is an interesting study that aimed to demonstrate that adipocyte-specific HMOX-1 gene therapy is a therapeutic approach for preventing the development of obesity-induced metabolic disease in an obese mice model. In general the manuscript is well written; please check throughout the text for grammar and spelling errors. Methods section and statistical analysis section are clearly described. The figures and tables are clear and useful. Please, if appropriate, include a statement in the conclusion paragraph on the future direction and possible clinical application of the results.

Author Response

Reviewer 3

Comments and Suggestions for Authors

This is an interesting study that aimed to demonstrate that adipocyte-specific HMOX-1 gene therapy is a therapeutic approach for preventing the development of obesity-induced metabolic disease in an obese mice model. In general the manuscript is well written; please check throughout the text for grammar and spelling errors. Methods section and statistical analysis section are clearly described. The figures and tables are clear and useful. Please, if appropriate, include a statement in the conclusion paragraph on the future direction and possible clinical application of the results. This is indicated in the last statement.

Submission Date

06 November 2019

Date of this review

17 Nov 2019 20:47:37

Reviewer 4 Report

The research group has a longstanding interest and focus on the cytoprotective mechanism of HO-1 in various dieaes models and made important contributins to the field. This work aims to further unravel cell type specific mechnisms of HO-1 in obesity by targeting adipose tissue using a lentiviral promoterspecific approach. Injection of a lentivirus carrying HO-1 under control of an AdipoQ promoter led to a number of beneficial effects in HFD-treated mice including reduced adipocyte hypertrophy, fibrosis, decreased mitochondrial respiration, increased levels of inflammatory adipokines, insulin resistance, vascular dysfunction and impaired heart mitochondrial signaling. The results shown support the conclusions in most cases, however, a few major and minor comments remain:

Major comments:

The authors conclude that ‚Ln-adipo-HO-1 administration mediated induction of HO-1 expression only in adipose tissue‘ - Their in vivo HO-1 targeting approach should also result in expression of HO-1 in subcutaneous as well as brown fat as the AdipoQ promoter use din the lentiviral construct is active also in these fat tissue and especially sc fat has important metabolic effects in obesoty  Have the authors measured HO-1 mRNA/protein in brown fat? As the authors report typical brown fat genes (Ucp1, Prdm16) – did HO-1 overexpression in sc and brown  fat impact expression of these genes? HO-1 protein levels in visceral fat of ln-Adipo-HO-1 and ln-GFP injected mice must be included. HO-1 affected body weight - Was food intake different between the three animal groups? The authors state that fat-specific HO-1 overexpression reduces HFD-triggered Insulin resistance, however, Insulin resistance (in vivo) has not been measured – rather, effects of HO-1 on Insulin resistance is inferred from phosphorylation status of Insulin signaling cascade, certainly important signal transduction pathways implicated in insulin signaling. Have the authors performed intraperitoneal Insulin tolerance tests or collected blood insulin levels?  Lines 386-391: Numbers for basal, maximal, uncouped and ATP-linked respiration must be reported in a separate (bar) graph comparing these values among the 3 cell lines including appropriate statistical evaluation. Display of OCR traces over time is not sufficient to conclude: ‚ATP turnover was significantly decreased in HO-1 ablated cells. The maximum respiration was also significantly (p<0.05) lower in HO1 deficient cells which was rescued after overexpression of HO-1. Mitochondria isolated from HO-1 ablated cells also showed lower coupling efficiency indicating a lower proportion of oxygen consumed to drive ATP synthesis compared with that driving proton leak which was rescued after overexpression of HO-1 (Figure 8) (p < 0.05)‘. Did HO-1 affect uncoupling efficiency? This is important information relating to the scheme shown in Figure 12, which proposes effects of HO-1 overexpression on heat production by beige and/or brown fat. Figure 8: The authors do not describe the method used for Twist1 silencing. This information must be included. Was Twist 1 silenced in preadipocytes or mature adipocytes? Successful knockdown of Twist1 mRNA and/or protein levels must be shown before concluding ‚TWIST1 expression regulates HO-1 and PGC-1α in cultured adipocyte cells‘. Figure 8: Maximal respiration rate must be calculated and reported as bar graph comapring ctrl- vs Twist1 knockdown. The subfigures should be labeled as done for Figure 7

Minor comments:

What is the difference between ‚ln-adipo-HO-1‘and ‚Lenti-adipo-HO-1‘ ? These terminologies are mixed throughout the manuscript Lines 517-519: Twist1 is not a cytokine – the sentence should be rephrased accordingly. At what age was HFD started and how long was the diet given? Line 98: When exactly were the two doses of lenti-HO-1 injected? Figure 7 G: It seems the color blocks indicating the individual respiration parameters are somewhat ‚shifted‘ – this should be corrected to accurately match with injection of compounds. Same issue with the black arrows. Can adipose HO-1 affect serum adiponectin levels in HFD fed mice (lenti-adipo-HO-1 versus lenti-GFP)? Lines 53-54: ‚Low levels of HO-1 are correlated with increased hip to waist ratio and insulin resistance[13]. – What ist the evidence that the cited reference suppports this statement? It seems that the reference[13] is wrong. Line 52-33: ‚Visceral fat of obese subjects show elevation of reactive oxygen species (ROS) [12] and diminished levels of the anti-oxidant gene; heme oxygenase-1 (HO-1).‘ – Same issue as comment 1… reference [12] does not support the statement. In addtion: Give a reference supporting the latter part of your statement. Figure 1. F-H – The ordering oft he subfigures do not match the figure legend. F) Systolic blood pressure, G) Body weight (BW), H) Oxygen consumption (VO2),

Author Response

Reviewer 4

Comments and Suggestions for Authors

The research group has a longstanding interest and focus on the cytoprotective mechanism of HO-1 in various dieaes models and made important contributins to the field. This work aims to further unravel cell type specific mechnisms of HO-1 in obesity by targeting adipose tissue using a lentiviral promoterspecific approach. Injection of a lentivirus carrying HO-1 under control of an AdipoQ promoter led to a number of beneficial effects in HFD-treated mice including reduced adipocyte hypertrophy, fibrosis, decreased mitochondrial respiration, increased levels of inflammatory adipokines, insulin resistance, vascular dysfunction and impaired heart mitochondrial signaling. The results shown support the conclusions in most cases, however, a few major and minor comments remain:

Major comments:

The authors conclude that ‚Ln-adipo-HO-1 administration mediated induction of HO-1 expression only in adipose tissue‘ - Their in vivo HO-1 targeting approach should also result in expression of HO-1 in subcutaneous as well as brown fat as the AdipoQ promoter use din the lentiviral construct is active also in these fat tissue and especially sc fat has important metabolic effects in obesity. We appreciate your interest in brown fat.  The paper’s focus on targeting white fat not brown fat.  Brown fat is well protected by its HO-1 levels.  All our measurements are on endogenous fat (visceral fat) using adiponectin promoter will result in expression of HO-1 in all fat not other organs.  Have the authors measured HO-1 mRNA/protein in brown fat? As the authors report typical brown fat genes (Ucp1, Prdm16) – did HO-1 overexpression in sc and brown  fat impact expression of these genes? HO-1 protein levels in visceral fat of ln-Adipo-HO-1 and [control contains GFP] ln-GFP injected mice must be included. HO-1 affected body weight - Was food intake different between the three animal groups? Food intake was similar in the three groups. The authors state that fat-specific HO-1 overexpression reduces HFD-triggered Insulin resistance, however, Insulin resistance (in vivo) has not been measured – rather, effects of HO-1 on Insulin resistance is inferred from phosphorylation status of Insulin signaling cascade, certainly important signal transduction pathways implicated in insulin signaling. We included the pertaining results.  Have the authors performed intraperitoneal Insulin tolerance tests or collected blood insulin levels?  We measured fasting glucose and insulin receptor phosphorylation which is an indicative of increase insulin sensitivity. Lines 386-391: Numbers for basal, maximal, uncouped and ATP-linked respiration must be reported in a separate (bar) graph comparing these values among the 3 cell lines including appropriate statistical evaluation. Display of OCR traces over time is not sufficient to conclude: ‚ATP turnover was significantly decreased in HO-1 ablated cells. The maximum respiration was also significantly (p<0.05) lower in HO1 deficient cells which was rescued after overexpression of HO-1. OCR is more than sufficient to describe oxygen consumption rats which we focused on and at the point for basal maximum.  However, uncoupled ATP-linked respiration amount cannot be measured.  Mitochondria isolated from HO-1 ablated cells also showed lower coupling efficiency indicating a lower proportion of oxygen consumed to drive ATP synthesis compared with that driving proton leak which was rescued after overexpression of HO-1 (Figure 8) (p < 0.05)‘. Did HO-1 affect uncoupling efficiency? [Yes, as UCP1 protein is elevated i.e, increases uncoupling efficiency] This is important information relating to the scheme shown in Figure 12, which proposes effects of HO-1 overexpression on heat production by beige and/or brown fat. Figure 8: The authors do not describe the method used for Twist1 silencing. The method for TWIST 1 silencing is included in the method section. It is written clearly in the legend on results of Figure 8. This information must be included. Was Twist 1 silenced in preadipocytes or mature adipocytes? Successful knockdown of Twist1 mRNA and/or protein levels must be shown before concluding ‚TWIST1 expression regulates HO-1 and PGC-1α in cultured adipocyte cells‘. OCR value has been given and its already written control vs Sh TWIST1 are labeled in Figure 7.  Figure 8: Maximal respiration rate must be calculated and reported as bar graph comapring ctrl- vs Twist1 knockdown. The subfigures should be labeled as done for Figure 7

Minor comments:

What is the difference between ‚ln-adipo-HO-1‘and ‚Lenti-adipo-HO-1‘ ? These terminologies are mixed throughout the manuscript Lines 517-519 Terminology corrected Lnv-adipo-HO-1 throughout the manuscript.: Twist1 is not a cytokine – the sentence should be rephrased accordingly Has been rephrased. At what age was HFD started and how long was the diet given? HFD diets started a 6 weeks for 23 weeks as indicated in the text. Line 98: When exactly were the two doses of lenti-HO-1 injected? Injected at 10 and 11 weeks. Figure 7 G: It seems the color blocks indicating the individual respiration parameters are somewhat ‚shifted‘ – this should be corrected to accurately match with injection of compounds. It has been corrected. Same issue with the black arrows. Can adipose HO-1 affect serum adiponectin levels in HFD fed mice (lenti-adipo-HO-1 versus lenti-GFP)? Lines 53-54: ‚Low levels of HO-1 are correlated with increased hip to waist ratio and insulin resistance[13]. References are corrected, low levels of HO-1 is increased in obesity and insulin resistance. – What ist the evidence that the cited reference suppports this statement? It seems that the reference [13] is wrong Line 52-33 citation is corrected and statement has been rewritten: ‚Visceral fat of obese subjects show elevation of reactive oxygen species (ROS) [12] and diminished levels of the anti-oxidant gene; heme oxygenase-1 (HO-1).‘ – Same issue as comment 1… reference [12] does not support the statement. In addtion: Give a reference supporting the latter part of your statement. Figure legend is corrected for the ordering.  We thank the reviewers for their constructive comments.  Figure 1. F-H – The ordering of the subfigures do not match the figure legend. F) Systolic blood pressure, G) Body weight (BW), H) Oxygen consumption (VO2),

Submission Date

06 November 2019

Date of this review

26 Nov 2019 14:17:24

Round 2

Reviewer 4 Report

The mansucript has improved; however,

1. in contrast to what the authors say in their cover letter, I do not see any changes to figure 7G - my concerns have not been addressed.

2. I disagree that individual mitochondrial parameters cannot be calculated from the OCR traces (e.g. basal or ATP-linked respiration, uncoupoling efficiency, etc) - details for calculations are given in the seahorse bioscience webpages for the mitochondrial stress test. These calculations should be included.

'OCR is more than sufficient to describe oxygen consumption rats which we focused on and at the point for basal maximum.' - In that case, the sentences spanning lines 286-388 must be removed.

3. Lines 108-109: '..BzATP 108 (200 μM) or nigericin (10 μM) was injected'- (i) what exactly was the purpose of injecting these compunds and (ii) where are these data?  

4. Were OCR values normalized against total protein or DNA content?

Author Response

The mansucript has improved; however,

in contrast to what the authors say in their cover letter, I do not see any changes to figure 7G - my concerns have not been addressed.

I disagree that individual mitochondrial parameters cannot be calculated from the OCR traces (e.g. basal or ATP-linked respiration, uncoupoling efficiency, etc) - details for calculations are given in the seahorse bioscience webpages for the mitochondrial stress test. These calculations should be included.

'OCR is more than sufficient to describe oxygen consumption rats which we focused on and at the point for basal maximum.' - In that case, the sentences spanning lines 286-388 must be removed.

Reply: 

 As per suggestions of reviewer , we have added basal or ATP-linked respiration, coupoling efficiency in the figure no 7 H,I,J

'OCR ---------286-388 has been removed now.

Q3. Lines 108-109: '..BzATP 108 (200 μM) or nigericin (10 μM) was injected'- (i) what exactly was the purpose of injecting these compunds and (ii) where are these data?

Reply:  

We agree to the reviewer and line 108-109 was wrongly typed which has been removed

Q4. Were OCR values normalized against total protein or DNA content?

Reply:  

OCR values normalized against total protein and it has been added on page no 3 in method sections.